# Prevalence of Fabry Disease in Patients on Dialysis in France

**DOI:** 10.3390/ijms251810104

**Published:** 2024-09-20

**Authors:** Florence Sens, Laure Guittard, Bertrand Knebelmann, Olivier Moranne, Gabriel Choukroun, Valérie de Précigout, Cécile Couchoud, Isabelle Deleruyelle, Léa Lancelot, Liên Tran Thi Phuong, Thomas Ghafari, Laurent Juillard, Dominique P. Germain

**Affiliations:** 1Service de Néphrologie et d’Explorations Fonctionnelles, Hôpital Edouard Herriot, Hospices Civils de Lyon, F-69003 Lyon, France; florence.sens@chu-lyon.fr (F.S.); laurent.juillard@univ-lyon1.fr (L.J.); 2UMR Inserm 1060, Université Claude Bernard Lyon 1, F-69621 Villeurbanne, France; 3Service Recherche et Epidémiologie Cliniques, Pôle Santé Publique, Hospices Civils de Lyon, F-69002 Lyon, France; laure.guittard@chu-lyon.fr (L.G.); isabelle.deleruyelle@chu-lyon.fr (I.D.); lea.lancelot@chu-lyon.fr (L.L.); 4Research on Healthcare Performance (RESHAPE), Inserm U1290, Université Claude Bernard Lyon, F-69373 Lyon, France; 5Service de Néphrologie, Hôpital Necker, Assistance Publique Hôpitaux de Paris, Université de Paris, F-75015 Paris, France; bertrand.knebelmann@aphp.fr; 6Service Néphrologie-Dialyse-Apherese, Hôpital Universitaire Caremeau, IDESP Université de Montpellier, F-30029 Nîmes, France; olivier.moranne@chu-nimes.fr; 7Service de Néphrologie, Médecine Interne, Dialyse et Transplantation, CHU Amiens, F-80054 Amiens, France; choukroun.gabriel@chu-amiens.fr; 8Service de Néphrologie, Hôpital Pellegrin, CHU Bordeaux, F-33076 Bordeaux, France; valerie.deprecigout@chu-bordeaux.fr; 9Coordination Nationale Réseau Epidémiologique et Information en Néphrologie, Agence de la Biomédecine, F-93212 Saint-Denis-La-Plaine, France; cecile.couchoud@biomedecine.fr; 10“Geneo” Referral Center for Fabry Disease and Lysosomal Diseases, MetabERN European Reference Network, F-92380 Garches, France; phuonglienvn282@gmail.com; 11Division of Medical Genetics, APHP—Paris Saclay University, F-92380 Garches, France; thomas.ghafari@aphp.fr; 12Division of Medical Genetics, University of Versailles, F-78180 Montigny, France

**Keywords:** Fabry disease, dialysis, screening, prevalence, genetic variants

## Abstract

Numerous prevalence studies on Fabry disease (FD, OMIM #301500) have been conducted in dialysis populations across the world with variable and controversial results. The FABRYDIAL study aimed to estimate the prevalence of FD in patients aged 18 to 74 years on chronic dialysis in France. This cross-sectional study was conducted in patients undergoing dialysis. One hundred and twenty-four dialysis centers participated. Patients with proven causes of nephropathy unrelated to FD were excluded. Alpha-galactosidase A activity was assayed in men, and both α-galactosidase A and lyso-Gb_3_ were assayed in women from dried blood spots. *GLA* gene sequencing was performed in case of abnormal values. If a variant was identified, a diagnosis validation committee was consulted for adjudication. Among the 6032 targeted patients, 3088 were included (73.6% of the eligible patients). Biochemical results were available for 2815 (1721 men and 1094 women). A genetic variant of *GLA* was identified in five patients: a benign c.937G>T/p.(Asp313Tyr) variant in two individuals, a likely benign c.427G>A/(p.Ala143Thr) variant, a likely benign c.416A>G/(p.Asn139Ser) variant, and a pathogenic c.1185dupG/p.Phe396Glyfs variant. Among the screened patients, the prevalence was 0.058% [0.010;0.328] in males, 0% [0.000;0.350] in females, and 0.035% [0.006;0.201] when both genders were pooled. Among all patients aged 18–74 years undergoing dialysis without a previously known cause of nephropathy unlinked to FD, the prevalence was 0.028% [0.006;0.121]. The prevalence of FD in a cohort of French dialysis patients was low. However, considering the prognostic impact of earlier diagnosis, signs of FD should be sought in patients with nephropathies of uncertain etiology.

## 1. Introduction

Fabry Disease (FD, OMIM #301500) is an X-linked lysosomal disease due to pathogenic variants in the *GLA* gene encoding for α-galactosidase A. It results in the accumulation of globotriaosylceramide (Gb_3_) and its deacetylated derivative (lyso-Gb_3_) in fluids and cellular lysosomes [1]. It is characterized by a wide phenotypic spectrum. In classic FD, patients typically exhibit neuropathic pain, angiokeratomas, hypohidrosis, gastrointestinal dysfunction, and *Cornea verticillata* from childhood or adolescence [1], but these early indicative signs are inconsistent [2,3]. Over the course of the disease, patients are susceptible to developing end-organ manifestations by the 4th or 5th decade of life, represented by heart failure and arrhythmias [4], end-stage-renal-disease (ESRD) [3,5,6], stroke [7], and premature death. Female patients are often symptomatic [8,9], although usually to a lesser extent than hemizygous males [10]. Since manifestations in adulthood are non-specific, FD is under-diagnosed. Screening for FD in high-risk groups is important for optimal management and family screening [3,10,11,12].

Many studies on the prevalence of FD in dialysis patients have been performed over the past twenty years [13,14,15,16,17,18,19,20,21,22,23,24,25,26,27,28,29,30,31,32,33,34,35,36,37,38]. Measured prevalence varied from 0 to 1.6% among males and 0 to 0.54% among females, with a systematic review suggesting a prevalence of 0.21% for males and 0.15% for females [39]. These differences could indicate a disparity in FD prevalence due to differences in healthcare system organization, leading to bias in patient selection. They may also be the consequence of methodological issues [40,41], such as an insufficient size of the populations screened [6,7,8,10,11,14,16,17,18], inaccurate screening tests [8,9,11,12,13,14,18,20,22], non-availability of confirmatory *GLA* variant analyses [16,17], and the absence of genotype/phenotype correlation to differentiate polymorphisms from pathogenic variants [39,42,43,44] since variants disclosed during a screening program may more likely be a mere coincidence and therefore benign than those disclosed in the course of a dedicated genetic outpatient clinic [45,46]. Therefore, the estimated prevalence of FD in many previous screening studies is questionable, and its generalizability to the population undergoing dialysis in France remains controversial.

The aim of the FABRYDIAL study was to estimate the prevalence of FD in a large cohort of French patients undergoing chronic dialysis aged 18 to 74 years. We also aimed to describe the phenotype of patients who harbor a *GLA* variant.

## 2. Results

### 2.1. Patients

Among the 6032 patients aged 18 to 74 years on chronic dialysis in the participating centers over the period considered, 714 were no longer treated at the centers when research staff visited for eligibility assessment, and 1121 met non-inclusion criteria, 89% of whom had a previously confirmed diagnosis of nephropathy unrelated to FD. A total of 3088 patients were included (73.6% of the 4197 eligible patients; 1888 men and 1200 women). Data from biochemical analyses were available for 2815 patients (1721 men and 1094 women); 97% (2719/2815) of samples were sent to the same laboratory (Figure 1). The median age was 63.5 years, and 61.1% of patients were men. In total, 97.7% of patients were treated with hemodialysis and 2.3% with peritoneal dialysis. The distribution of causal nephropathies was close to that of the French registry of dialysis patients when considering the exclusion criteria (Table 1).

### 2.2. Biochemical Screenings and Genetic Analyses

*GLA* gene analysis was performed for 91 patients; a genetic variant was found in five of them (four males and one female) (Table 2).

Review and adjudication of the five patients identified with a GLA variant were conducted by the multidisciplinary DVC (Table 2).

#### 2.2.1. Patient 1: No FD

The patient was a 66-year-old man with systemic hypertension, atrial fibrillation, and cardiac failure. At the age of 61, he presented with a sclerodermic renal crisis confirmed by biopsy, and dialysis was initiated. In 2016, he had benefited from an α-GalA assay that was lowered to 1.7 µmol/L/h (ref. ≥ 2.6), lyso-Gb3 was 1.8 ng/mL (ref. < 1.8), *GLA* sequencing found a missense variant c.416A>G (p.Asn139Ser), initially suspected to be associated with a late-onset cardiac variant [47]. However, the pathogenicity of this variant was subsequently challenged [48]. The screening was redone in 2017–2018: α-GalA was normal (4.11 µmol/L/h). The p.Asn139Ser variant was found 27 times in ExAC and 34 times in the gnomAD exome database. Both SIFT and Polyphen2 predicted a benign variant. The normal α-Gal A value on DBS in this male patient ruled out FD, while the history was not suggestive. We suggest that the p.Asn139Ser variant is a likely benign *GLA* variant not associated with FD [48].

#### 2.2.2. Patient 2: No FD

This 68-year-old woman began dialysis at the age of 26, one year after membranoproliferative glomerulonephritis was diagnosed by renal biopsy. She reported fatigue, acroparesthesias (attributed to beta-2-microglobulin amyloidosis), abdominal pain, and constipation. α-GalA was moderately lowered (0.99 µmol/L/h), lyso-Gb_3_ was within normal values (1.18 ng/mL), *GLA* gene sequencing identified a heterozygous benign variant c.937G>T (p.Asp313Tyr) [42,49]. The diagnosis of Fabry disease was not confirmed.

#### 2.2.3. Patient 3: No FD

This 47-year-old man was followed for chronic kidney disease since he was 40 and began dialysis at the age of 45 in the context of hepato-renal syndrome due to ethylic cirrhosis. Since 2016, he also had concentric hypertrophic cardiomyopathy (interventricular septum thickness: 14 mm). No renal biopsy had been performed. α-GalA was mildly lowered but with a significantly elevated residual level (0.86 µmol/L/h). *GLA* gene sequencing identified a non-pathogenic hemizygous variant c.937G>T (p.Asp313Tyr) [42,49,50]. FD diagnosis was not confirmed.

#### 2.2.4. Patient 4: No FD

This 61-year-old man presented with cobalt pulmonary fibrosis and required a bipulmonary transplant at the age of 59, the consequences of which were marked by six months in intensive care units and dialysis at the age of 60. No renal biopsy had been performed. α-GalA was only slightly lowered (1.04 µmol/L/h). *GLA* gene sequencing identified a hemizygous variant c.427G>A (p.Ala143Thr) of unknown significance, which has been extensively discussed in the literature with several lines of evidence in favor of a likely benign variant [42,43,44,50,51,52,53]. FD diagnosis was not confirmed.

#### 2.2.5. Patient 5: Confirmed FD

A 63-year-old man was diagnosed with chronic kidney disease at age 54, and hemodialysis was initiated at age 61. No renal biopsy had been performed, and his nephropathy was of undetermined origin. Family history disclosed a medical history of cardio-renal disease in his mother and heart failure in two brothers who died at ages 42 and 59. The patient had no children. His medical records reported concentric hypertrophic cardiomyopathy of unknown cause: interventricular septum thickness: 14 mm). According to his nephrologist, the patient had no early signs suggestive of FD. However, careful clinical examination by a senior geneticist (DPG) disclosed chronic fatigue, hypohidrosis, heat intolerance, chronic diarrhea, hypoacousia, and tinnitus signs that can be found in association with FD. Additionally, hypohidrosis and angiokeratoma in the periumbilical and genital areas, which are symptoms highly suggestive of FD, were also evident. α-GalA activity was dramatically decreased (0.1 µmol/L/h), and lyso-Gb_3_ was highly elevated (117.8 ng/mL). A class 5 pathogenic frameshift variant (c.1185dupG; p.Phe396Glyfs) was detected in the *GLA* gene, thereby definitely confirming the diagnosis of classic FD. The patient was initiated on enzyme replacement therapy (agalsidase beta), on which he remained for 5 years before passing away.

### 2.3. Prevalence of Fabry Disease in Dialysis Patients

FD was diagnosed in only one male and no female patients. Among the patients included in the FABRYDIAL study and screened (2815 patients), the prevalence of FD was 0.058% [0.010;0.328] in males, 0% [0.000;0.350] in females, and 0.035% [0.006;0.201] regardless of gender. Among eligible included patients (3088 patients), the prevalence was 0.053% [0.011;0.307] in males, 0% [0.000;0.319] in females, and 0.032% [0.007;0.188] regardless of gender. We considered that (i) all eligible patients (n = 4197) had the same prevalence as the 2815 included patients, i.e., 0.035%, (ii) the 1121 patients with an exclusion criteria linked to their nephropathy had a prevalence of 0%, and (iii) the 714 patients who were no longer present in the dialysis center at the time of the study had to be excluded from the prevalence calculation because we could not form a hypothesis. The prevalence among the target population (i.e., patients aged 18 to 74 years undergoing chronic dialysis without a previously known cause of nephropathy unlinked to FD) is therefore (0.035 × 4197 + 0 × 1121)/(4197 + 1121), i.e., 0.028% [0.006;0.121]; an estimation by sex was not possible because the gender of non-included patients was unknown for regulatory reasons.

## 3. Discussion

The FABRYDIAL study aimed to estimate the prevalence of FD in patients aged 18 to 74 years undergoing chronic dialysis in France. A total of 124 dialysis centers participated in the study, and 3088 patients were included. Biological samples were available for 2815 patients, namely the α-Gal A activity in men and both α-Gal A and lyso-Gb_3_ in women. Among the cases with first-tier abnormal result(s) in whom *GLA* gene sequencing was performed, five patients were found to have a *GLA* genetic variant. After analysis by the DVC, only one male was considered to have FD. Among the patients included in the study and screened (2815 patients), the prevalence of FD was 0.058% [0.010;0.328] in males, 0% [0.000;0.350] in females, and 0.035% [0.006;0.201] in whatever the gender. Among all the patients aged 18 to 74 years undergoing chronic dialysis, the prevalence was estimated at 0.016%.

Many studies have been conducted to estimate the prevalence of FD in ESRD patients (Table 3) [6,7,8,9,10,11,12,13,14,15,16,17,18,19,20,21,22,23,24], but most have limitations [39,40,41]. Many lacked power (<1000 patients) [6,7,8,10,11,14,16,17,18,37], and most excluded women [6,7,10,15,16,17,19,24] or screened them only for α-GalA activity [8,9,11,12,13,14,18,20,21,22], possibly leading to false negatives [3,16,40]. The present study offers an approach that is both economically viable and scientifically valid, using a sensitive biological screening in women and a diagnostic algorithm consistent with recent data [30,53,54]. Another limitation of previously published studies is the frequent lack of distinction between benign and pathogenic variants of *GLA* [41,42,43,44,45,50,51], which can lead to overestimating the prevalence by up to 50% [25]. Finally, in countries in which there is no free access to dialysis, FD prevalence could be biased.

According to a meta-analysis, the prevalence of FD in dialysis is around 0.21% in men and 0.15% in women [25], much higher than the prevalence measured in the present study. When limiting the scope to studies in European populations of at least 1000 patients, all were carried out prior to 2008, and the prevalence they measured varied from 0.26% to 0.42% in men [3,12,14]. More recently published studies unrelated to Europe have observed similar results, except for the Japanese J-FAST study [28] and the Australian aCQuiRE study [38], which both measured a prevalence of 0.02% [28,38], close to that found in the present study.

Among the hypotheses that may explain the low prevalence found in the present study was the fact that a previously known diagnosis of FD would have been an exclusion criterion. However, investigators had to request the inclusion of patients for whom FD was confirmed, and an analysis of the data of the National Dialysis Registry (Registre Epidémiologie et Information en Néphrologie, REIN [55]) did not identify any patients with previously diagnosed FD among the screened patients, suggesting that no patient with a known diagnosis of FD was excluded. The low prevalence could also be due to a lack of sensitivity, as measuring α-GalA on dried blood carries a risk of false negatives [16]; however, in our study, the α-GalA threshold was set high, so this risk was negligible in men. In women, the risk of having both α-GalA and lyso-Gb_3_ in the normal ranges along with ESRD due to FD is also likely negligible [3,30,42,43]. However, cases of later-onset Fabry disease with high residual enzyme activity in heterozygote females in association with a coincident kidney disease leading to ESRD could have been missed. Alternatively, the prevalence in this study could reflect reality more accurately than in previous studies. Our approach aggregated phenotypic, biological, and genetic data, thereby avoiding over-diagnosing FD. As an example, thorough phenotype-genotype correlation allowed us to consider a patient with a likely benign variant (p.Asn139Ser), initially suspected to be associated with the later-onset cardiac phenotype of FD, as non-FD.

In the only diagnosed case of FD in this study, suggestive early signs and family history had been overlooked prior to the study. However, after examination, the FD-expert physician (DPG) identified a suggestive familial and personal history. The diagnosis led to the initiation of specific therapy and family screening [56]. This case emphasizes the importance of searching for a suggestive family history [56,57] and the need for nephrologists to be trained in recognizing the physical signs of FD, in particular when the etiology of the nephropathy is unknown.

This study has several limitations, including the absence of participation of all French dialysis centers, the restriction to dialyzed patients rather than to all ESRD patients, a high rate (26.4%) of non-included eligible patients, and the lack of data related to such patients, especially regarding their gender. More importantly, diagnosis in female Fabry patients may have been missed since a recent large retrospective study conducted on 827 patients (374 males and 453 females), all with a causative variant in the *GLA* gene, showed that while 100% of male patients had an α-GalA activity lower than the reference value, regardless of the phenotype, more than 70% of female patients had normal enzyme activity, both with the classic variant and with late-onset variants. Those female patients would, therefore, escape diagnosis if the enzymatic test alone was used [58]. Similarly, 42% of females with a pathogenic variant in *GLA* had normal plasma lyso-Gb3 values with a significant difference between subjects with the classic variant compared to patients with a later-onset form of the disease [45,58]. In the case of classic Fabry disease, 83% of female patients had pathological values of plasma LysoGb3, meaning that using this test for diagnosis, up to 17% of female patients could, in theory, have been missed [58]. However, previous studies in female patients with Fabry disease have shown a strong correlation between low alpha-galactosidase activity (in association with a highly skewed X inactivation profile) and progression to end-stage renal disease, thereby highly reducing the risk of a false negative in our study population [8,9].

The strengths of the present study advocate for the scientific relevance of its results. The screening was conducted in a country where the national healthcare system covers the costs of dialysis, reducing patient selection bias. It relied on data from a nationwide dialysis database (REIN) [55,59]. Patients’ screening was limited to patients dialyzed on a single date, according to the strict definition of prevalence. The protocol requested the inclusion of previously diagnosed FD patients in order to measure the prevalence of FD and not of “undiagnosed-FD”. The diagnostic sensitivity was enhanced thanks to a systematic dosage of lyso-Gb_3_ in addition to the α-Gal assay in women. Finally, the DVC reduced the risk of over-diagnosis of FD through the exclusion of non-pathogenic *GLA* variants [41,42,43,44,45,50,51].

In conclusion, the FABRYDIAL study aimed at estimating the prevalence of FD in patients aged 18 to 74 years undergoing chronic dialysis in France. Among the 2815 patients with available biological data, one patient was diagnosed with FD, leading to an estimated prevalence of 0.036% [0.006;0.201], 0.058% in men [0.01;0.328], and 0% in women [0;0.35]. When considering patients who were not included due to a previously known cause of nephropathy unlinked to FD, the prevalence in this population was lower, 0.028% [0.006;0.121]. The only FD-diagnosed patient presented with a highly suggestive history and typical signs that should have evoked FD prior to initiating dialysis [60]. Nephrologists should be aware of FD and look for its early signs when caring for patients with nephropathy of uncertain etiology since earlier diagnosis and treatment of FD are associated with better outcomes [1,10,60,61,62,63,64,65,66,67,68,69].

## 4. Patients and Methods

### 4.1. Context

A standardized screening approach was proposed to dialysis centers, targeting patients aged 18 to 74 years undergoing chronic dialysis in order to identify and confirm FD patients. Patients were not eligible if the cause of their nephropathy was known and unrelated to FD, which was restricted to the following causes: polycystic kidney disease, type 1 diabetes or biopsy-proven IgA nephropathy, and membranous glomerulonephritis or ANCA-associated vasculitis. All patients confirmed non-objection before participation.

### 4.2. Study Design and Ethics Statement

This cross-sectional study targeted patients undergoing chronic dialysis in five geographical areas of France (Ile-de-France, Rhône-Alpes, Aquitaine, Picardie, and Gard). All dialysis centers were invited to participate in the study by local study coordinators; a total of 124 centers were included. The study received approval from the French National Data Protection Agency (Reference Methodology MR-003—Commission Nationale Informatique et Libertés, number 17–002). It was classified as non-interventional by the Review Board of the Regional Ethics Committee (2 December 2015). All eligible patients were offered to participate in the study. They were given a written explanatory document, and their non-objection was recorded. The study was in agreement with the Helsinki Declaration of 1975, as revised in 2013.

### 4.3. Subjects

Patients aged 18 to 74 years from all participating centers undergoing chronic dialysis during the week of 20 November 2017, were evaluated for eligibility. Eligibility criteria included a previous or scheduled FD screening test and a non-objection to participate in the study. Non-eligibility criteria included proven nephropathy unrelated to FD (same criteria as those of the FD screening campaign), the absence of health insurance coverage, or guardianship or tutelage. Reasons for non-inclusion of eligible patients could not be collected for regulatory reasons.

Data collection was facilitated by extracting data from the National Dialysis Registry (REIN [38]). All of the extracted data were systematically checked in the patient medical files.

### 4.4. Biochemical Methods

#### 4.4.1. Blood Spot Tests

In the setting of routine care, two dried blood spots (DBS) were collected from patients on a filter paper (Whatman number 903 paper, Whatman, Middlesex, UK). The cards were air-dried, kept at room temperature during shipment, and subsequently stored at 4 °C. Diagnostic tests were performed at ARCHIMED Life Science GmbH (Vienna, Austria).

#### 4.4.2. Diagnostic Algorithm

Screening for FD was carried out by measuring the activity of α-GalA in men and both α-GalA activity and lyso-Gb_3_ levels in women, using DBS samples. Samples were analyzed in replicates, and borderline results were re-tested before a 2nd card was requested and/or genetic testing was performed. In patients with decreased α-GalA activity (<1.2 µmol/L/h), increased Lyso-Gb_3_ (>3.5 ng/mL), or both, *GLA* sequencing was performed.

#### 4.4.3. α-GalA Activity Determination

α-GalA activity was determined using a validated and accredited fluorimetric method. DBS samples were added to 70-well microlitre plates and incubated overnight at 37 °C with 4-methylumbelliferyl-α-D-galactopyranoside as the substrate and N-acetyl-D-galactosamine as an inhibitor for α-N-acetylgalactosaminidase. Each plate contained an internal standard for quality control purposes. The reaction was stopped with 300 µL ethylenediamine (pH 11.3) [70].

#### 4.4.4. Lyso-Gb_3_ Determination

Lyso-Gb_3_ concentrations were measured by highly-sensitive electrospray ionization liquid chromatography–tandem mass spectrometry (ESI LC-MS/MS) using a modified method based on Gold et al. [71] as previously described [72,73]. A 7-point serum calibrator, an internal standard for lyso-Gb_3_ quantification (analytic range from 0–120 ng/mL; lower limit of quantification: 1.5 ng/mL), and three calibrator levels for quality control were used.

### 4.5. Genetic Analysis

The *GLA* mutations were identified by Sanger DNA sequencing using standard methods [46].

### 4.6. Diagnosis Validation Committee

A multidisciplinary diagnosis validation committee (DVC) was in charge of confirming or excluding FD diagnosis for patients with a *GLA* genetic variant. The committee included the nephrologist in charge of the patient, two senior nephrologists (FS and BK), and a senior geneticist with recognized expertise in FD (DPG). The diagnosis was confirmed in the case of the following associations: (1) Biochemical abnormalities (α-GalA activity <1.2 µmol/L/h and/or lyso-Gb3 >3.5 ng/mL), (2) A likely pathogenic (class 4) or pathogenic (class 5) genetic variant and (3) A compatible phenotype. Variants were assessed using gnomAD, ClinVar, and HGMD2018.2 [42,58,74,75]. Unpublished variants were interpreted according to the ACMG/AMP 2015 guidelines [74].

### 4.7. Statistical Analysis

Descriptive statistics were used for clinical characteristics and laboratory parameters. Categorical variables were expressed as absolute counts and percentages (%). Continuous variables of normal distribution were summarized by the mean and standard deviation (SD), and continuous variables of non-normal distribution by the median and interquartile range (IQR). Group comparisons were performed using Chi-squared tests for categorical variables and Student’s *t*-tests for continuous variables; where distributional assumptions for these tests were violated, the Fisher’s exact and Mann–Whitney–Wilcoxon tests, respectively, were used.

The prevalence of FD was determined by the proportion of patients with a confirmed diagnosis. The prevalence among included and screened patients was the main endpoint. The prevalence of FD was then estimated among the total population of patients aged 18 to 74, based on the following hypotheses: (1) Non-included eligible patients, and patients with an absence of health insurance or guardianship or tutelage, have a prevalence of FD identical to that of screened patients; (2) Patients excluded due to a renal diagnosis unrelated to FD do not have FD.

Statistical significance was defined as *p*-value < 0.05. All statistical analyses were performed using the SAS statistical software version 9.3 (SAS Institute, Cary, NC, USA).

## Figures and Tables

**Figure 1 ijms-25-10104-f001:**
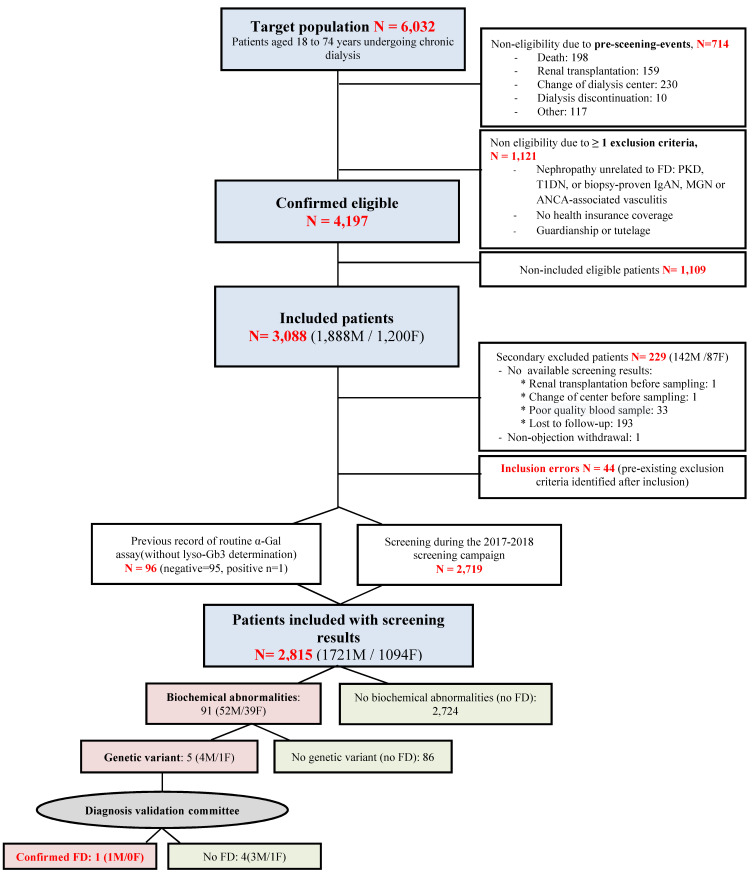
Flow diagram of the FABRYDIAL study.

**Table 1 ijms-25-10104-t001:** Characteristics of patients included in the FABRYDIAL study.

	Patients Included with Screening Results
	*n* = 2815
Sex (male), *n* (%)	1721 (61.1)
Age (years), median [IQR]	63.5 [54.2–69.5]
Diagnosis of ESRD, *n* (%)	
Systemic hypertension	654 (23.3)
Other vascular nephropathies	64 (2.3)
Type 2 diabetic nephropathy	768 (27.3)
Glomerulonephritis	375 (13.3)
Tubulo-interstitial impairment and toxicities	231 (8.2)
Non-renal urinary tract diseases	202 (7.2)
Hereditary diseases (except PKD)	37 (1.3)
Other causes	128 (4.5)
Unknown	356 (12.6)
Previous α-galactosidase A assay	96 (3.4)
Negative	95 (3.37)
Positive	1 (0.03)
Dialysis modality, *n* (%), (6 missing data)	
Hemodialysis	2745 (97.7)
Peritoneal dialysis	64 (2.3)
Main home in the geographical areas of the study, n (%), (22 missing data)	2729 (97.7)

Abbreviations: ESRD: End-stage renal disease, IQR: Interquartile range, PKD: Polycystic kidney disease.

**Table 2 ijms-25-10104-t002:** Clinical, biological, and genetic data from patients identified with a variant identified in *GLA*.

Sex	Age	Familial History	Kidney Disease	Age at ESRD	Early Signs	Cardiac Dysfunction	Comorbidities	α-GalA Activity	Lyso-Gb3	Gene Variant	Final Diagnosis
M	66	0	Sclerodermic renal crisis	61	No	Altered left ventricular ejection fraction HF	Hypertension, Atrial fibrillation, Depression	4.1 (previously 1.7 in 2016)	Previously 1.8 in 2016	c.416A>G (SNV)	Negative
F	68	0	Membrano-proliferative GN	26	Acroparesthesia (late onset), abdominal pains	No	None	0.99	1.2	c.937G>T (SNV)	Negative
M	47	0	Hepato-renal syndrome	45	No	Ventricular hypertrophy without HF	Ethyl cirrhosis	0.86	-	c.937G>T (SNV)	Negative
M	61	0	Calcineurin Inhibitors toxicity	60	No	No	Cobalt fibrosis, Bipulmonary transplantation	1.04	-	c.427G>A (SNV)	Negative
M	63	First-degree premature deaths (1, 42 and 59 y.o.)	Unknown	61	Hypohidrosis, heat intolerance, diarrhea, tinnitus, angiokeratoma, hypoacousia	Hypertrophic cardiomyopathy at 38 y.o. + HF	Atrial fibrillation	0.15	117.8	c.1185dupG (Frameshift mutation)	Fabry disease

Abbreviations: α-GalA: α-galactosidase A, ESRD: End-stage renal disease, F: Female, GN: Glomerulonephritis, HF: Heart failure, Lyso-GB3: Lyso-globotriaosylcéramide, M: Male, SNV: Single nucleotide variant, y.o.: years old.

**Table 3 ijms-25-10104-t003:** Summary of previously published screening studies for Fabry disease in dialysis patients.

Author/Year	Country	Population	No of Patients	Male (%)	Screening Male	Screening Female	No Fabry Male/ Female	Frequency (%) Male/ Female
Nakao/2003 [14]	Japan	He	514	100	Plasma α-GAL > WBC α-GAL leuco > Gn	None	6/NA	1.17/NA
Linthorst/ 2003 [15]	The Netherlands	He/PD	508	100	WBC α-GAL > Gn	None	1/NA	0.22/NA
Bekri/2004 [16]	France	He	106	56	WBC α-GAL > Gn	WBC α-GAL > Gn	1/0	1.69/0.00
Kotanko/2004 [17]	Austria	He/PD	2480	61	Plasma α-GAL > WBC α-GAL leuco > Gn	Plasma α-GAL > WBC α-GAL leuco > Gn	4/0	0.26/0.00
Ichinose/2005 [18]	Japan	He/PD	450	100	Plasma α-GAL > WBC α-GAL leuco > Gn	None	1/NA	0.22/NA
Tanaka/2005 [19]	Japan	He	696	58	Plasma α-GAL > WBC α-GAL leuco > Gn	Plasma α-GAL > WBC α-GAL leuco > Gn	4/1	1.00/0.34
Merta/2007 [20]	Czech	He	3370	45	Plasma α-GAL > WBC α-GAL leuco > Gn	AGAL blood spot > AGAL leuco > Gn	4/3	0.26/0.16
Terryn/2008 [21]	Belgium	He	1284	18	Plasma α-GAL > WBC α-GAL leuco > Gn	Plasma α-GAL > WBC α-GAL leuco > Gn	½	0.42/0.19
De Schoen-makere/2008 [22]	Belgium	Gr	673	41	Plasma α-GAL > Gn	Plasma α-GAL > Gn	1/0	0.36/0.00
Kleinert/2009 [23]	Austria	Gr	1306	100	Plasma α-GAL > Gn	None	2/NA	0.15/NA
Andrade/ 2008 [24]	Canada	CKD/PD/He/Gr	141/59/159/138	100	Plasma α-GAL > WBC α-GAL	None	0/NA	0.00/NA
Porsch/2009 [25]	Brazil	He	558	100	Plasma α-GAL	None	2/NA	0.36/NA
Gaspar/2010 [26]	Spain	He	911	60	Plasma α-GAL > Gn	Plasma α-GAL > Gn	4/4	0.55/0.54
Maruyama/ 2013 [27]	Japan	He/PD	1453	100	Plasma α-GAL + Lyso-Gb3 > Gn	None	4/NA	0.28/NA
Saito/2015 [28]	Japan	He/PD	8547	63	Plasma α-GAL > WBC α-GAL > Gn	Plasma α-GAL > WBC α-GAL leuco > Gn	2/0	0.04/0.00
Sodre/2017 [29]	Brazil	He	36,442 (Qnaire) --> 8087 (Bio)	60	Questionnaire → Plasma α-GAL + Gn	Questionnaire --> Plasma α-GAL + Gn	26/45	0.11/0.31
Moiseev/2019 [30]	Russia	He	5572	64	Plasma α-GAL > Gn + Gb3	Plasma α-GAL --> Gn + GB3	19/1	0.53/0.05
Yalin/2019 [31]	Turkey	He/Gr	5477 (1652 He/3822 Gr)	63	Plasma α-GAL > Gn	Gn	17 (2 He/15 Gr)	0.31 (0.12He; 0.39Gr)
Frabasil/2019 [32]	Argentina	He/PD	9604	100	α-GAL > Gn	None	22/NA	0.23/NA
Jahan/2020 [33]	Australia		526		DBS α-GAL > Gn	DBS α-GAL > Gn	0/0	0.00/0.00
Alhemyadi/ 2020 [34]	Saudi Arabia		619		DBS α-GAL > WBC α-GAL > Gn	DBS α-GAL > WBC α-GAL > Gn	0/3	0.00/0.48
Nagata/2022 [35]	Japan	He/PD	2122 Including 1703		Plasma α-GAL > WBC α-GAL > Gn	None	1/NA	0.06/NA
Vigneau/2022 [36]	Western France	CKD stage 5D/T	819 including 242 CKD stage 5D	100	DBS α-GAL	NA	0/NA	0.00/NA
Mallett/2022 [37]	Australia	CKD stage 1-5D/T	2992	58	DBS α-GAL + Lyso-Gb3 > Gn	DBS α-GAL + Lyso-Gb3 > Gn	5/1	0.3/0.08
Cho/2024 [38]	South Korea	CKD stage 1–5	897 including 279 CKD stage 5D	45	α-GAL + Lyso-Gb3 > Gn	α-GAL + Lyso-Gb3 > Gn	0/1	0.00/0.2
Present study	France	He/PD	2815	61	α-GAL > Gn	α-GAL + Lyso-GB3 > Gn	1/0	0.06/0.00

Abbreviations: α-GalA: α-galactosidase A, CKD: Chronic Kidney Disease, He: hemodialysis, Lyso-Gb3: Lyso-globotriaosylceramide, PD: peritoneal dialysis.

## Data Availability

The main data underlying this publication are available in the article. In accordance with French regulations, additional data, since they are not necessary to answer the authors’ scientific question, namely the prevalence of Fabry disease, cannot be shared.

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
