# Peer review of "Prevalence of Fabry Disease in Patients on Dialysis in France"

_ijms, 2024, doi:10.3390/ijms251810104_

Round 1

Reviewer 1 Report

Comments and Suggestions for Authors

Thank you to the authors for submitting this manuscript describing a large cohort of ~3000 French dialysis patients who were screened for Fabry Disease in a regimented and structured program. It is well written and communicates the concepts very well. I have some queries.

1) In regard to the patient consenting, please clarify if this was active consent (ie, each patient signed a consent form) or passive consent (ie, opt-out)?

2) In Table 3 there are perhaps a small number of recent studies that have been missed, most particularly the aCQuiRE study (https://pubmed.ncbi.nlm.nih.gov/35505287/) which actually had a very similar methodology/study design, cohort size, and findings compared to this study. 

Author Response

1) In regard to the patient consenting, please clarify if this was active consent (ie, each patient signed a consent form) or passive consent (ie, opt-out)?

Response: the study was qualified as non-interventional (MR-003) by the Review Board of the regional ethics committee. As a consequence, all eligible patients were offered to participate in the study. They were given a written explanatory document and their passive consent (“non-objection” in the manuscript) was recorded. One sentence has been modified in the manuscript for more clarity (All patients confirmed non-objection before participation.)

2) In Table 3 there are perhaps a small number of recent studies that have been missed, most particularly the aCQuiRE study

Response: we have followed this interesting suggestion and retrieved additional recent screening studies (n= 7) which have been added to both table 3 and the reference list (references #33 to #3). In particular, the aCQuiRE study, which methodology was indeed very similar to the FABRYDIAL study, has been added and mentioned in the text of the manuscript.

Reviewer 2 Report

Comments and Suggestions for Authors

The paper titled "Prevalence of Fabry Disease in Patients on Dialysis in France" by Florence Sens should be published. It demonstrates that in France, which could be seen as representative of many developed countries, the population of undiagnosed Fabry disease patients undergoing dialysis is relatively low. However, there are some issues that need to be addressed.

In the summary the authors state that:

“Among the 6,032 targeted patients, 3,088 were included (73.6% of eligible patient). Biochemical results were available for 2,815 (1,721 men and 1,094 women). A genetic variant of GLA was identified in 5 patients: c.937G>T or p.(D313Y) in 4 individuals and a pathogenic frameshift variant  c.1185dupG/p.Phe396Glyfs in a 60-year-old male. “

This is not clear because the authors describe 5 patients and 4 genetic variants in the results

Patient 1 c.416A> G (p.Asn139Ser)

Patient 2 c.937G>T 216 (p.Asp313Tyr)

Patient 3 c.937G>T (p.D313Y)

Patient 4 c.427G>A (p.A143T)

Patient 5 (c.1185dupG; p.Phe396Glyfs).

The way variants are presented should be consistent.( p.Asp313Tyr, p.D313Y for example)

In Table 1, they examined screening before and during the 2017-2018 campaign. However, they combined the patients' results. I would have preferred to present the biochemical results separately for the screenings before and during the 2017-2018 campaign. This approach would have made it clearer to see how the diagnosis of Fabry disease is improving in France.

In the results, the figures calculated to report the prevalence of Fabry disease in dialysis patients should be double-checked because some are incorrect.

In general, the references are unbalanced in favour of previous papers by the same authors. References to relevant papers by other groups should be considered.

 "GLA" should be in italics whenever it refers to the gene.

Author Response

1) In the summary the authors state that: “Among the 6,032 targeted patients, 3,088 were included (73.6% of eligible patient). Biochemical results were available for 2,815 (1,721 men and 1,094 women). A genetic variant of GLA was identified in 5 patients: c.937G>T or p.(D313Y) in 4 individuals and a pathogenic frameshift variant c.1185dupG/p.Phe396Glyfs in a 60-year old male. “

This is not clear because the authors describe 5 patients and 4 genetic variants in the results

Patient 1 c.416A> G (p.Asn139Ser)

Patient 2 c.937G>T 216 (p.Asp313Tyr)

Patient 3 c.937G>T (p.D313Y)

Patient 4 c.427G>A (p.A143T)

Patient 5 (c.1185dupG; p.Phe396Glyfs).

Response: we very much thank the reviewer for this critical comment. For an unknown reason a mistake had occurred in the abstract. The data in the main manuscript are accurate. The abstract has been modified accordingly.

2) The way variants are presented should be consistent: (p.Asp313Tyr, p.D313Y for example)

Response: we agree with the reviewer and the three-letter nomenclature has now been homogenously adopted for all variants in both the abstract and all along the manuscript.

3) In Table 1, they examined screening before and during the 2017-2018 campaign. However, they combined the patients' results. I would have preferred to present the biochemical results separately for the screenings before and during the 2017-2018 campaign. This approach would have made it clearer to see how the diagnosis of Fabry disease is improving in France.

Response: No screening study was conducted before the 2017-2018 campaign. For a small number of cases (96 out of 2815 = 3.4%), a medical record of previous a-galactosidase assay was noted, but without data on preanalytical steps and lyso-Gb3 values.

While inclusion of those 96 (mostly negative) patients may have slightly lowered the prevalence (0.035 instead of 0.037%), all 2,815 patients benefited from exactly the same standardized, controlled, structured screening procedure within the FABRYDIAL study.

The word “screening” has been withdrawn and replaced by “assay” for those cases to avoid any confusion since we have no additional data allowing to compare diagnostic efficacy over time in France.

In addition, to take into account the reviewer’s comment we have added three lines in table 1 highlighting the previous record of an a-Gal A assay in 96 patients (with negative results for 95 of them and the (false) positive for 1 patient - presented in depth in the result section). We feel that this add-on improves completeness and accuracy of the results.

4) In the results, the figures calculated to report the prevalence of Fabry disease in dialysis patients should be double-checked because some are incorrect.

Response : we have double-checked  the figures and corrected them when needed. We thank the reviewer for his/her expert advices.

5) In general, the references are unbalanced in favour of previous papers by the same authors. References to relevant papers by other groups should be considered.

Response: the point made by the reviewer was well taken and two references from our group have been withdrawn (Caudron et al 2015; Germain and Jurca-Simina 2018) while twenty additional references from a variety of colleagues(refs #33-39, #41, #43, #57, #60-69, #75) have been added to better acknowledge their contribution and balance the reference list.

6) "GLA" should be in italics whenever it refers to the gene.

Response: GLA has been italicized all along the manuscript when refering to the gene.